# The impact of team psychological safety on employee innovative performance a study with communication behavior as a mediator variable

Hao Jin[1¤a], Yan Peng[2¤b]*

1 School of Management, Lyceum of The Philippines University-Batangas, Batangas, Philippines, 2 School of Economics and Management, Guangzhou Vocational University of Science and Technology, Guangzhou, China

¤a Current address: Zhengzhou City, Henan Province, China
¤b Current address: Guangzhou, Guangdong Province, China
* yanni_pp@hotmail.com

**Data Availability Statement:** All relevant data are within the manuscript and its Supporting Information files.

## Abstract

This study aims to delve into the impact of team psychological safety on employee innovative performance and to analyze the mediating role of communication behavior in this process. Given the central role of innovation in the sustained competitiveness of enterprises, understanding how team psychological safety promotes employee innovation through communication behavior is of significant theoretical and practical importance. The study employs a structural equation model (SEM) to analyze survey data from 580 employees across various high-tech enterprises. The results indicate that the three dimensions of team psychological safety—team collaboration and understanding, team information sharing, and team give-and-take balance—have a significant positive impact on employee innovative performance. Moreover, communication behavior plays a significant mediating role between team psychological safety and employee innovative performance. This study reveals the positive influence of team psychological safety on employee innovative performance through communication behavior, emphasizing the importance of building a positive team psychological safety environment to stimulate employee innovation potential. It provides empirical evidence for enterprise managers to optimize team communication strategies and enhance employee innovative performance.

## 1. Introduction

In the rapidly changing business environment of today, innovation has become the key to gaining a competitive advantage for businesses. Employees, as the main executors of innovative activities, directly affect the company's innovation capabilities and market position with their innovative performance. Team psychological safety, as an important psychological state that promotes knowledge sharing and team collaboration, is considered a key environmental

**Funding:** This work was funded and supported by the key research and development program of Hebei Province "Applied Research on Improving the Quality of Obstetrical Anesthesia Based on Deep Learning" (Project Number: 22377766D). The sponsors had no role in the study design, data collection and analysis, decision to publish, or preparation of the manuscript.

**Competing interests:** The authors have declared that no competing interests exist.

factor in stimulating the innovative potential of employees [1]. A secure team environment can encourage members to propose new ideas, try new methods, and learn from failures, thereby driving organizational innovation. Although the importance of team psychological safety has been widely recognized, the specific mechanism by which it affects employee innovative performance is not yet fully clear. Moreover, the mediating role of communication behavior between team psychological safety and employee innovative performance is relatively understudied [2]. Therefore, exploring how team psychological safety affects employee innovative performance through communication behavior can not only enrich the theoretical framework of team innovation but also provide practical strategies for organizations to enhance innovative performance [3].

Existing literature has confirmed the positive correlation between team psychological safety and employee innovative performance and has preliminarily explored the role of communication behavior in the team innovation process [4]. However, most studies have focused on the separate impacts of team psychological safety or communication behavior, lacking in-depth analysis of their interaction and comprehensive impact on innovative performance. This study delves into the specific mechanism by which team psychological safety affects employee innovative performance, especially the mediating role of communication behavior [5] By applying structural equation modeling to specifically analyze the direct impact of each dimension of team psychological safety on employee innovative performance and how communication behavior acts as a mediating variable, the aim is to provide a more detailed empirical analysis for the existing literature.

## 2. Theoretical background and assumptions

### 2.1 The relationship between team psychological safety and employee innovative performance

In recent years, research in the field of organizational behavior has increasingly focused on the concept of team psychological safety [6]. Team psychological safety is defined as a team climate where members feel safe enough to take risks, propose new ideas, and try new strategies without worrying about potential negative consequences [7]. Employee innovative performance involves the innovation capabilities and outcomes demonstrated by employees within an organization, including but not limited to the proposal of innovative ideas, the implementation of innovation projects, and the improvement of existing processes. Scholars have indicated that there is a significant positive correlation between team psychological safety and employee innovative performance. These studies provide preliminary evidence for understanding how team psychological safety promotes innovation [8]. However, the existing literature lacks detailed descriptions of how the various dimensions of team psychological safety specifically affect employee innovative performance, and there is a relative lack of in-depth exploration of this mechanism [9]. This study selects team collaboration and understanding, team information sharing, and team give-and-take balance as key dimensions of team psychological safety for in-depth analysis [10]. Team collaboration and understanding is at the core of team psychological safety, emphasizing collaborative cooperation among members, which has a catalytic effect on stimulating employees' innovative thinking. Team information sharing, as a key pathway to promoting the flow of knowledge and the generation of innovative inspiration, helps to break down information silos [11]. Team give-and-take balance is closely related to job satisfaction and engagement, playing a crucial role in creating a positive working atmosphere [12]. Based on this, the study proposes the following hypotheses:

H1.1: Team collaboration and understanding have a significant positive impact on employee innovative performance.

H1.2: Team information sharing has a significant positive impact on employee innovative performance.

H1.3: Team give-and-take balance has a significant positive impact on employee innovative performance.

## 2.2 The relationship between team psychological safety and communication behavior

In recent years, team psychological safety has become a focal point of research in the field of organizational psychology, describing a psychological state where team members feel safe, willing to share knowledge, propose new ideas without fear of negative consequences [13]. Communication behavior is regarded as the core process of information exchange and collaboration within a team, affecting the team's decision-making, problem-solving, and innovation capabilities [14]. Scholars have pointed out that team psychological safety is a key factor in promoting team communication and collaboration [15]. It not only encourages team members to openly exchange opinions and feedback but also provides a supportive environment for team members, thereby promoting the team's innovative capabilities. Although existing literature has recognized the importance of team psychological safety for communication behavior, there is a lack of in-depth exploration of how team psychological safety affects communication behavior through different dimensions [16]. This study aims to deepen this understanding by examining the three dimensions of team collaboration and understanding, team information sharing, and team give-and-take balance to reveal how they collectively affect communication behavior. Therefore, the study proposes the following hypotheses:

H2.1: Team collaboration and understanding positively promote team members' communication behavior.

H2.2: Team information sharing positively promotes team members' communication behavior.

H2.3: Team give-and-take balance positively promotes team members' communication behavior.

## 2.3 The relationship between communication behavior and employee innovative performance

In the context of organizational innovation, communication behavior forms the foundation of organizational interaction, promoting not only the flow of information and knowledge sharing but also stimulating employees' innovative thinking and problem-solving abilities [17]. Employee innovative performance refers to the innovation-related behaviors and outcomes demonstrated by individuals within an organization, including proposing innovative ideas and implementing innovation projects [18] Studies have shown that communication behavior is crucial for organizational innovation, providing a social foundation for innovation by promoting the exchange of ideas and integration of knowledge among team members [19]. Moreover, an open and supportive communication environment can increase employees' willingness to share innovative ideas, increasing the likelihood that these ideas will be adopted and implemented [20] Although existing literature has emphasized the role of communication behavior in promoting employee innovative performance, further research is needed on how

communication behavior specifically affects individual levels of innovation motivation and behavior and its mechanisms in different organizational contexts [21]. This study aims to explore how communication behavior stimulates employees' internal motivation and innovative spirit by enhancing their sense of participation and belonging [22]. Therefore, the study proposes the following hypothesis:

H3: There is a significant positive relationship between communication behavior and employee innovative performance.

## 2.4 The mediating role of communication behavior

In organizational behavior research, team psychological safety is considered a key factor affecting employee innovative performance [23]. The role of communication behavior in this relationship, especially as a mediating variable, is significant for understanding how team psychological safety affects employee innovative performance. The mediation effect analysis framework proposed by Baron and Kenny (1986) provides a theoretical foundation for studying communication behavior [24]. According to this framework, communication behavior is selected as a mediating variable to explore its potential role between team psychological safety and employee innovative performance. Although existing studies have confirmed the positive impact of team psychological safety on employee innovative performance, there is a lack of in-depth exploration of the role of communication behavior as a mediating variable [25]. This study aims to clarify the mediating effect of communication behavior between team psychological safety and employee innovative performance through empirical analysis, providing new insights for organizations on how to enhance team innovation capabilities by optimizing communication strategies [26]. Therefore, the study proposes the following hypothesis:

H4: Communication behavior plays a significant mediating role between team psychological safety and employee innovative performance.

## 3. Research methodology

This study employs a Structural Equation Model (SEM) to elucidate the influence of team psychological safety on employee innovative performance and to scrutinize the mediating role of communication behavior [27]. The SEM was selected for its strengths in handling complex variable relationships and its capability to provide a comprehensive assessment of the model's overall fit [28]. This approach allows for the precise quantification of direct and indirect effects among the independent variable, mediating variable, and dependent variable [29]. It offers a rigorous analytical framework for understanding how team psychological safety impacts employee innovative performance through communication behavior, as depicted in (Fig 1).

### 3.1 Questionnaire design and data collection

To ensure the content validity of the scale, all items in this study refer to relevant literature from both domestic and international sources [30]. Additionally, the scale was refined through consultation and interviews with corporate executives to better align with the actual context in China. All items were measured using a 5-point Likert scale, where 5 indicates "very consistent" and 1 indicates "very inconsistent." The study collected data through a survey questionnaire, targeting high-tech enterprises known for their innovation-centric focus, high knowledge intensity, and emphasis on employee innovative performance [31]. To enhance the generalizability of the research findings, more than 40 companies across various cities, including Beijing, Shanghai, Tianjin, Wuhan, and Guangzhou, were selected as the sample. In the

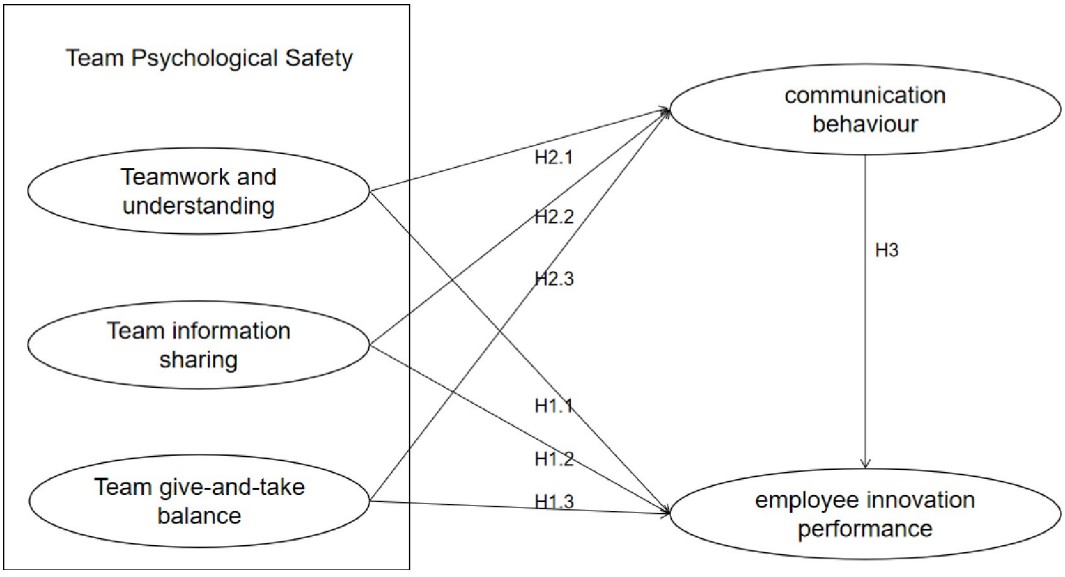

**Fig 1. Research hypothesis path model.**

statistical process, a total of 600 questionnaires were distributed, with 580 valid responses successfully collected, achieving an effective recovery rate of 96.7%. Specific descriptive analysis is presented in (Table 1).

## 3.2 Scale items and exploratory analysis

As shown in (Table 2), all variables have Cronbach's alpha coefficients higher than 0.7, indicating that the scale has a high level of reliability. This result is consistent with the scale reliability standards proposed by [32], ensuring the stability and reliability of the research data. The

**Table 1. Demographic characteristics (N = 580).**

| Variable | Mean | standard deviation | variance | Options | Frequency | Percent |
|---|---|---|---|---|---|---|
| Gender | 1.54 | 0.02 | 0.25 | Male | 265 | 45.7 |
| | | | | Female | 315 | 54.3 |
| Age | 3.17 | 0.34 | 0.66 | 18–24 years old | 16 | 2.8 |
| | | | | 25–34 years old | 102 | 17.6 |
| | | | | 35–54 years old | 232 | 40.0 |
| | | | | 55 years old and above | 230 | 39.7 |
| Edu | 1.75 | 0.03 | 0.39 | Associate's or Bachelor's degree | 204 | 35.2 |
| | | | | Master's degree | 316 | 54.5 |
| | | | | Doctorate or higher | 60 | 10.3 |
| Work Experience | 2.81 | 0.40 | 0.96 | 1–2 years | 82 | 14.1 |
| | | | | 3–5 years | 90 | 15.5 |
| | | | | 6–8 years | 266 | 45.9 |
| | | | | 9–10 years | 142 | 24.5 |
| Employee Positions | 2.54 | 0.05 | 1.12 | Frontline Staff | 144 | 24.8 |
| | | | | Middle Management | 122 | 21.0 |
| | | | | Senior Management | 170 | 29.3 |
| | | | | Leadership Level | 144 | 24.8 |

**Table 2. Scale question items and their reliability test results.**

| Measurement variable | Questionnaire | Source | Factor Loading | CR | AVE | alpha value |
|---|---|---|---|---|---|---|
| Teamwork and understanding | Team members are easily understood and accepted by other colleagues. | [34] | 0.803 | 0.851 | 0.652 | 0.818 |
| | Immature ideas are valued in the team. | | 0.823 | | | |
| | Team members focus on building common understanding in their communication | | 0.797 | | | |
| Team information sharing | Team members usually share information within the team rather than exclusively. | [35] | 0.807 | 0.856 | 0.664 | 0.822 |
| | Team members focus on teamwork and sharing. | | 0.838 | | | |
| | Team members communicate with each other about work-related issues. | | 0.799 | | | |
| Team give-and-take balance | There is both give and take in teams. | [35] | 0.819 | 0.843 | 0.642 | 0.826 |
| | Team members are fairly rewarded for their efforts and contributions within the team | | 0.790 | | | |
| | Team members are able to give their time and effort as well as experience a sense of satisfaction and achievement in their work | | 0.794 | | | |
| Communication behaviour | Team members better understand the organisation's goals through effective communication. | [36] | 0.831 | 0.853 | 0.660 | 0.832 |
| | Communication helps team members share innovative ideas more actively. | | 0.810 | | | |
| | In collaboration, communication behaviours motivate team members to produce more creative solutions. | | 0.796 | | | |
| Employee innovation performance | Employees often come up with innovative ideas and solutions. | [37] | 0.764 | 0.834 | 0.626 | 0.833 |
| | Employees demonstrate higher levels of innovation and output in their teams. | | 0.811 | | | |
| | Members' innovative performance has a positive impact on the team as a whole. | | 0.798 | | | |

structural validity of the scale includes both convergent validity and discriminant validity. Following the recommendations of, we assessed the standardized factor loadings of the variables, which ranged from 0.764 to 0.838, indicating good convergent validity of the scale. Additionally, the Combined Reliability (CR) and Average Variance Extracted (AVE) of the scale are both greater than their respective acceptable values of 0.7 and 0.5, further confirming the scale's convergent validity [33].

According to the discriminant validity test proposed by Fornell and Larcker, this paper calculated the square root of AVE for each variable and the correlation coefficient between variables [38]. As shown in (Table 2), the square root of AVE for each column (in bold and italics) is greater than the correlation coefficient between variables, indicating that the scale has good discriminant validity.As shown in (Table 3).

## 4 Results of the study

### 4.1 Model fitness test

This paper uses structural equation modelling to test the direct and mediating effects between variables, using the software AMOS24.Before testing the hypotheses [39], it is necessary to test the overall fitness of the model.The test results are: χ2/df = 1.950 (less than 3), RMSEA = 0.041 (less than 0.08), GFI = 0.967 (greater than 0.9), AGFI = 0.95 (greater than 0.9), CFI = 0.981

**Table 3. Results of distinguished validity analysis.**

| | 1 | 2 | 3 | 4 | 5 |
|---|---|---|---|---|---|
| 1 Teamwork and understanding | *0.807* | | | | |
| 2 Team information sharing | 0.41 | *0.815* | | | |
| 3 Team give-and-take balance | 0.37 | 0.371 | *0.801* | | |
| 4 Communication behaviour | 0.35 | 0.332 | 0.459 | *0.812* | |
| 5 Employee innovation performance | 0.44 | 0.426 | 0.491 | 0.465 | *0.791* |

**Table 4. Path analysis results.**

| Hypothesis | Path | B-value | T-value | Results |
|---|---|---|---|---|
| H1.1 | Teamwork and understanding→Employee innovation performance | 0.247*** | 4.378 | Adjuvant |
| H1.2 | Team information sharing→Employee innovation performance | 0.189*** | 3.372 | Adjuvant |
| H1.3 | Team give-and-take balance→Employee innovation performance | 0.292*** | 5.240 | Adjuvant |
| H2.1 | Teamwork and understanding→Communication behaviour | 0.188** | 3.182 | Adjuvant |
| H2.2 | Team information sharing→Communication behaviour | 0.132* | 2.220 | Adjuvant |
| H2.3 | Team give-and-take balance→Communication behaviour | 0.412*** | 5.240 | Adjuvant |
| H3 | Communication behaviour→Employee innovation performance | 0.240*** | 7.517 | Adjuvant |

Note:*, ** and *** denote significance levels of 0.05, 0.01 and 0.001, respectively, the same below

(greater than 0.9), NFI = 0.962 (greater than 0.9), IFI = 0.981 (greater than 0.9) [40]. All indicators met the measurement criteria, indicating a good overall fit of the model.

## 4.2 Hypothesis testing

**4.2.1 Direct path test.** (Table 4), presents the results of the path analysis of the research model. There is a significant positive effect between teamwork and understanding, team information sharing and team give-and-take balance and employee innovation performance, and hypotheses H1.1, H1.2 and H1.3 are tested [41]. There is a significant positive effect between teamwork and understanding, team information sharing and team give-and-take balance and communication behaviour, hypotheses H2.1, H2.2 and H2.3 are tested. The path coefficient between communication behaviour and employee innovation performance reaches the significance level of 0.001 and hypothesis H3 is supported.

**4.2.2 Test of mediating role.** In order to test the mediating role of tacit knowledge sharing, the bootstrap method that does not require the assumption of normal distribution is used. AMOS software provides the corresponding calculation function [42], and the sample number of bootstrap is chosen to be 5,000, and the test results are shown in (Table 5), with a confidence interval of 95%.

As can be seen in (Table 5), the p-values of all three paths are less than 0.05 and the confidence intervals do not include 0, indicating a significant mediating role. This means that communication behaviour plays the role of a mediating variable in the relationship between teamwork and understanding, team information sharing, team give-and-take balance and innovation performance [43]. Therefore, hypothesis H4 is partially tested.

## 5 Discussion

The results of this study reveal the significant positive impact of team psychological safety on employee innovative performance, and communication behavior plays an important mediating role in this process. These findings are consistent with theoretical framework, which suggests that team psychological safety can promote innovative behavior among team members.

**Table 5. Results of the mediation effect test.**

| Path | B-value | Lower value | Upper value | P-value |
|---|---|---|---|---|
| Teamwork and understanding→Communication behaviour→Employee innovation performance | 0.045** | 0.017 | 0.09 | 0.002 |
| Team information sharing→Communication behaviour→Employee innovation performance | 0.032* | 0.004 | 0.074 | 0.026 |
| Team give-and-take balance→Communication behaviour→Employee innovation performance | 0.099*** | 0.051 | 0.156 | 0.000 |

Specifically, the three dimensions of team psychological safety—team collaboration and understanding, team information sharing, and team give-and-take balance—all enhance communication behavior, thereby improving employee innovative performance [44]. Hypotheses H1.1, H1.2, and H1.3 proposed the positive influence of the three dimensions of team psychological safety on employee innovative performance, and the results indicate that these hypotheses are supported. Furthermore, hypotheses H2.1, H2.2, H2.3, and H4 proposed the mediating role of communication behavior between team psychological safety and employee innovative performance, which has also been verified. From a theoretical perspective, this study provides new insights into team innovation theory by distinguishing different dimensions of team psychological safety and examining how they affect employee innovative performance through communication behavior [45]. On the practical level, the results of this study suggest that business managers should pay attention to the construction of team psychological safety, especially in creating a work environment that supports communication and collaboration. By enhancing a culture of collaboration within the team, encouraging information sharing, and ensuring a fair give-and-take mechanism, organizations can effectively improve employee innovative performance. The study also provides a specific mechanism by which team psychological safety affects employee innovative performance, namely, the mediating role of communication behavior.

## 6 Conclusion

This study explored the impact of team psychological safety on employee innovative performance through structural equation modeling, with a particular focus on the mediating role of communication behavior. The findings reveal that team psychological safety significantly and positively affects employee innovative performance, with communication behavior playing a pivotal mediating role in this process. Specifically, the dimensions of teamwork and understanding, team information sharing, and team give-and-take balance all enhance employee innovative performance by promoting communication behavior. By distinguishing different dimensions of team psychological safety, this study provides a more nuanced theoretical perspective on how team psychological safety affects employee innovative performance and uncovers the mediating role of communication behavior between team psychological safety and employee innovative performance. Future research could investigate how different types of communication behaviors affect the relationship between team psychological safety and employee innovative performance.

## Supporting information

**S1 Data.**
(XLSX)

## Acknowledgments

Upon the completion of this study, we wish to express our deepest gratitude to the individuals and institutions that have made valuable contributions to the project. Our first thanks go to all the corporate employees who participated in the survey; without their active participation and honest feedback, this research would not have been possible.

We would like to extend our special thanks to YP, His expertise and valuable time were crucial in ensuring the quality of our research.

We confirm that we have obtained consent from all individuals mentioned in the acknowledgement, and they are aware and agree to be acknowledged in this section of the study.

## Author Contributions

**Conceptualization:** Yan Peng.

**Methodology:** Yan Peng.

**Writing – original draft:** Hao Jin.

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
