## [Decision Letter · Decision Letter 0]

1 May 2024

PONE-D-23-42648The effect of psychological safety on innovation performance: communication behaviour as a mediating variablePLOS ONE

Dear Dr. jin,

Thank you for submitting your manuscript to PLOS ONE. After careful consideration, we feel that it has merit but does not fully meet PLOS ONE’s publication criteria as it currently stands. Therefore, we invite you to submit a revised version of the manuscript that addresses the points raised during the review process.

We look forward to receiving your revised manuscript.

Kind regards,

Ricardo Limongi

Academic Editor

PLOS ONE

Journal Requirements:

2. We note that your Data Availability Statement is currently as follows: "All relevant data are within the manuscript and its Supporting Information files."

3. Please ensure that you refer to Figure 1 in your text as, if accepted, production will need this reference to link the reader to the figure

Reviewers' comments:

Reviewer's Responses to Questions

**Comments to the Author**

1. Is the manuscript technically sound, and do the data support the conclusions?

Reviewer #1: Yes

Reviewer #2: Partly

Reviewer #3: Yes

2. Has the statistical analysis been performed appropriately and rigorously? 

Reviewer #1: Yes

Reviewer #2: I Don't Know

Reviewer #3: Yes

3. Have the authors made all data underlying the findings in their manuscript fully available?

Reviewer #1: Yes

Reviewer #2: No

Reviewer #3: No

4. Is the manuscript presented in an intelligible fashion and written in standard English?

Reviewer #1: Yes

Reviewer #2: Yes

Reviewer #3: Yes

5. Review Comments to the Author

Reviewer #1: Abstract

The research problem, objectives, and significance of your study was clearly stated. A brief literature review on the main concepts of team psychological safety and innovation performance was also provided.

However, there are some grammatical and punctuation errors in the introduction, such as "fully understoodAndersson". You should proofread your abstract carefully and correct these mistakes.

Literature Review

The literature review is comprehensive and well-structured, covering relevant studies on team psychological safety, communication behavior, and innovation performance. the review also effectively highlights the gaps in existing research and justifies the need for the current study.

Under 2.1 The relationship between team psychological safety and employee innovative performance:

"In past research, scholars have begun to focus on the relationship between team psychological safety and employee innovation performance. However, there is still a certain research gap in terms of how specific dimensions of team psychological safety affect innovation performance and the interrelationship between them." These emphatic statement(s) above must be referenced

However, I also think that it would be great if the two sections (Introduction and Literature Review) could be merged, to form one section.

Research Methods

The methodology section provides detailed information on the sample selection, data collection procedures, and scale design, enhancing the transparency and replicability of the study. Rigorous statistical analysis methods, including structural equation modeling and reliability testing, are appropriately described. I suggest that the information about the ethical approval and consent procedures of the study could be also be included.

The study; "The effect of psychological safety on innovation performance: communication

behaviour as a mediating variable" presents a well-structured study focusing on an important topic in organizational psychology, with clear objectives, a thorough literature review, robust methodology, and sound statistical analysis. The findings contribute to understanding the intricate relationship between team psychological safety, communication behavior, and employee innovation performance.

Reviewer #2: After an initial assessment, I do not consider your paper to meet the journal’s criteria for publication. I have particular concerns about the lack of a clear research question; the robustness of the methods and analysis; the soundness and basis of the conclusions; the contribution to the literature; and the clarity of the narrative and expression. I am not confident that the issues identified could be resolved with even major revisions.

Reviewer #3: Strengths of the Manuscript

1. Empirical Rigor: The manuscript employs structural equation modeling, a robust statistical technique, which provides a comprehensive analysis of the complex relationships between psychological safety, communication behavior, and innovation performance. This approach adds significant empirical strength to the study.

2. Innovative Mediating Variable: Introducing communication behavior as a mediating variable is a notable strength. It enriches the discourse on psychological safety's role in innovation, highlighting the nuanced pathways through which team dynamics affect performance outcomes.

3. Comprehensive Data Collection: The collection of 580 valid samples from high-tech enterprises across multiple major cities enhances the generalizability of the findings. This large and diverse sample size lends credibility and robustness to the study's conclusions.

4. Multidimensional Analysis: The manuscript successfully identifies and elaborates on multiple dimensions of psychological safety—team collaboration and understanding, information sharing, and give-and-take balance. This multidimensional approach allows for a deeper understanding of the constructs at play.

5. Practical Implications: The study provides specific management strategies to maximize employee innovation performance, making it a theoretical contribution and a practical guide for business managers looking to foster a conducive environment for innovation.

Weaknesses of the Manuscript

1. Theoretical Contributions: The manuscript tends to rely heavily on existing theories and models rather than offering substantial new theoretical insights or frameworks. This might limit its appeal to academic audiences seeking innovative theoretical contributions.

2. Literature Integration: The discussion sometimes needs to fully integrate and contrast its findings with the broader body of existing research. A more thorough comparison with other studies could strengthen the manuscript's positioning within the current scientific debate.

3. Depth of Mediating Analysis: While introducing communication behavior as a mediator is a strength, the analysis could benefit from a deeper exploration of why and how this mediation occurs. Providing more detailed psychological or organizational behavior theories could enrich the explanation.

4. Operational Definitions: Some constructs, particularly 'team give-and-take balance,' could be better operationalized. Clarifying how these terms are measured and their relevance to psychological safety could improve the manuscript's clarity and impact.

5. Demographic Considerations: The manuscript mentions the demographic breakdown of the sample but does not explore how demographic variables (e.g., age, gender, educational background) might influence the relationships studied. This oversight might need to be clarified in the data.

6. Methodological Assumptions: The reliance on self-reported data through questionnaires raises concerns about bias, such as social desirability or respondent fatigue. Although the manuscript addresses common method variance, it could further discuss these limitations and their potential impact on the findings.

7. Data Analysis Transparency: More transparency in the statistical analysis, particularly in handling the structural equation modeling, would bolster the manuscript's credibility. Detailed descriptions of model fit indices, assumption checks, and validation measures would be beneficial.

8. Cross-Cultural Validity: Given that the study is conducted within high-tech enterprises in specific geographic locations, the generalizability of the findings to other cultural contexts or industries could be clearer. Discussing the cross-cultural applicability of the results could enhance the manuscript's relevance.

9. Recommendations for Practice: While the manuscript offers practical advice for managers, these recommendations could be expanded to include specific, actionable strategies or case study examples to illustrate the successful implementation of the proposed ideas.

10. Future Research Directions: The manuscript could more explicitly outline areas for future research, such as longitudinal studies to track changes over time or experimental designs to test causality. This would not only enhance the manuscript but also guide subsequent research efforts.

Suggestions for Improvement

- Enhancing the theoretical framework by integrating additional perspectives or proposing a new model based on the findings could provide a stronger theoretical contribution.

- Expanding the literature review to include a broader range of studies and more critically contrasting them with the current findings.

- Providing clearer operational definitions and more detailed explanations of the constructs and their interrelations.

- Incorporating demographic analyses to explore the influence of these variables on the main study relationships.

- Discuss methodological limitations more thoroughly and suggest ways to mitigate potential biases.

6. PLOS authors have the option to publish the peer review history of their article (what does this mean?). If published, this will include your full peer review and any attached files.

Reviewer #1: **Yes: **Andreas Ndapewa Frans

Reviewer #2: **Yes: **Aidin Salamzadeh

Reviewer #3: No

---

## [Author Response · Author response to Decision Letter 0]

16 Jun 2024

Response to reviewers

In response to the three questions posed by the editor, here are possible answers:

1. Concerning compliance with PLOS ONE style requirements and file naming requirements:

 Our file naming also follows journal guidelines to ensure clarity and consistency.

2. On the submission of data:

 We understand that we are not required to submit the entire dataset if only a portion of the data is used in the reporting study. We have selected data for submission that are directly relevant to the findings of the study.

3. With regard to the reference to figure 1 in the text:

 We thank the editorial team for their guidance and have made the necessary adjustments to the manuscript in accordance with PLOS ONE submission requirements. We are confident that these adjustments will aid in the review process and enable us to present our research more clearly to the academic community.

Comments to the author

1. Technical soundness and data to support conclusions

Our study design was technically sound and rigorous experimental manipulations were performed. We ensured that the experiments had appropriate control groups, sufficient number of replications and sample size. All conclusions are based on the data presented and have been carefully analysed. We believe that the data strongly support our research hypotheses and conclusions.

2. Rigour of statistical analyses:

Yes, we have carried out appropriate and rigorous statistical analyses. We have used advanced statistical methods to process the data and ensured that the analysis process has followed the standards recognised by the scientific community. All analyses were conducted by professional statisticians to ensure the accuracy and reliability of the results.

3. Provision of basic data

In accordance with the PLOS data policy, we have provided the underlying data for all study results. These data include, but are not limited to, the data points behind the mean, median, and variance measures. We have provided these data as supporting information for the manuscript and deposited them in a public repository where necessary. Restrictions on any data sharing, such as participant privacy or use of third-party data, have been described in our data availability statement.

4. English writing and clarity: 

Our manuscripts are written in standard English and have been thoroughly language proofread and edited for clarity and comprehensibility prior to submission. We have corrected any typographical or grammatical errors to meet PLOS ONE's language requirements.

5. Comments to the author: 

Reviewer comments are critical to improving the quality of the paper and adapting it to the journal's publication standards. Below are the responses to each reviewer's comments and suggestions for improvement:

Reviewer #1's response and suggestions for improvement:

1. Grammatical and punctuation errors:

 - We thank the reviewers for pointing out grammatical and punctuation errors in the abstracts. We will thoroughly proofread the abstracts to ensure that all text is accurate.

2. Introduction and literature review combined:

 - We understand the reviewers' suggestion and will consider combining the introduction and literature review into a more coherent section to improve the flow and clarity of the paper.

3. Ethical approval and consent procedures in research methodology:

 - We agree with the reviewers' suggestion to add information about the ethical approval and consent process of the study in the methodology section to enhance the transparency of the study.

### Reviewer #2 responses and suggestions for improvement:

1. Robustness of research questions, methods and analyses:

 - We will revisit the research questions to ensure that they are clear and specific. At the same time, we will conduct a further review of the methodology and analyses to enhance their robustness.

2. Reasonableness and basis for conclusions:

 - We will ensure that conclusions are based on adequate data analysis and provide a clear rationale.

3. Contribution to the literature:

 - We will explore in more depth the contribution of this study to the existing literature and how it fills the gaps in existing research in the discussion section.

4. Clarity of narrative and expression:

 - We review the entire paper for language and presentation to ensure that the narrative is clear and logical.

### Reviewer #3 responses and suggestions for improvement:

1. Theoretical contributions:

 - We thank the reviewers for their recognition. We will endeavour to present new insights or models within the theoretical framework to enhance the theoretical contribution of the paper.

2. Integration of literature:

 - We will broaden the scope of the literature review to more critically compare current findings with other studies to strengthen the paper's position in the academic debate.

3. Depth of brokering analysis:

 - We will provide more detailed theoretical support by exploring in greater depth why and how communication behaviour acts as a mediating variable.

4. Operational definitions:

 - We will provide clearer operational definitions, especially for key concepts such as the "team give-and-take balance" and their relevance to psychological safety.

5. Demographic considerations:

 - We will explore how demographic variables affect research relationships and consider these factors in data analysis.

6. Methodological assumptions:

 - We will discuss the limitations of relying on self-reported data and suggest ways to mitigate potential bias.

7. Transparency in data analysis:

 - We will provide a more detailed description of the statistical analyses, including model fit indices and hypothesis checking.

8. Cross-cultural effectiveness:

 - We will discuss the cross-cultural applicability of the findings and consider their generalisability in different cultural contexts.

9. Recommendations for practice:

 - We will expand the Practice Recommendations section to include specific operational strategies or case study examples.

10. Directions for future research:

 - We will outline areas for future research more clearly in the discussion section to provide guidance for subsequent research.

---

## [Editor Report · Decision Letter 1]

21 Jun 2024

The Impact of Team Psychological Safety on Employee Innovative Performance A Study with Communication Behavior as a Mediator Variable

PONE-D-23-42648R1

Dear Dr. jin,

We’re pleased to inform you that your manuscript has been judged scientifically suitable for publication and will be formally accepted for publication once it meets all outstanding technical requirements.

Kind regards,

Ricardo Limongi

Academic Editor

PLOS ONE

---

## [Editor Report · Acceptance letter]

15 Jul 2024

PONE-D-23-42648R1 

PLOS ONE

Dear Dr. Peng, 

I'm pleased to inform you that your manuscript has been deemed suitable for publication in PLOS ONE. Congratulations! Your manuscript is now being handed over to our production team.

Kind regards, 

on behalf of

Professor Ricardo Limongi 

Academic Editor

PLOS ONE